# Learning-based Support Estimation in Sublinear Time

**Talya Eden**[*], **Piotr Indyk, Shyam Narayanan, Ronitt Rubinfeld & Sandeep Silwal**
Computer Science and Artificial Intelligence Lab
Massachusetts Institute of Technology
Cambridge, MA 02139, USA
`{teden,indyk,shyamsn,ronitt,silwal}@mit.edu`

**Tal Wagner**
Microsoft Research
Redmond, WA 98052, USA
`talw@mit.edu`

## Abstract

We consider the problem of estimating the number of distinct elements in a large data set (or, equivalently, the support size of the distribution induced by the data set) from a random sample of its elements. The problem occurs in many applications, including biology, genomics, computer systems and linguistics. A line of research spanning the last decade resulted in algorithms that estimate the support up to $\pm\varepsilon n$ from a sample of size $O(\log^2(1/\varepsilon) \cdot n/\log n)$, where $n$ is the data set size. Unfortunately, this bound is known to be tight, limiting further improvements to the complexity of this problem. In this paper we consider estimation algorithms augmented with a machine-learning-based predictor that, given any element, returns an estimation of its frequency. We show that if the predictor is correct up to a constant approximation factor, then the sample complexity can be reduced significantly, to

$$\log(1/\varepsilon) \cdot n^{1-\Theta(1/\log(1/\varepsilon))}.$$

We evaluate the proposed algorithms on a collection of data sets, using the neural-network based estimators from Hsu et al, ICLR'19 as predictors. Our experiments demonstrate substantial (up to 3x) improvements in the estimation accuracy compared to the state of the art algorithm.

## 1 Introduction

Estimating the support size of a distribution from random samples is a fundamental problem with applications in many domains. In biology, it is used to estimate the number of distinct species from experiments (Fisher et al., 1943); in genomics to estimate the number of distinct protein encoding regions (Zou et al., 2016); in computer systems to approximate the number of distinct blocks on a disk drive (Harnik et al., 2016), etc. The problem has also applications in linguistics, query optimization in databases, and other fields.

Because of its wide applicability, the problem has received plenty of attention in multiple fields[1], including statistics and theoretical computer science, starting with the seminal works of Good and Turing Good (1953) and Fisher et al. (1943). A more recent line of research pursued over the last decade (Raskhodnikova et al., 2009; Valiant & Valiant, 2011; 2013; Wu & Yang, 2019) focused on the following formulation of the problem: given access to independent samples from a distribution

---

[*]Authors listed in alphabetical order

[1]A partial bibliography from 2007 contains over 900 references. It is available at
`https://courses.cit.cornell.edu/jab18/bibliography.html`.

$\mathcal{P}$ over a discrete domain $\{0, \ldots, n-1\}$ whose minimum non-zero mass[2] is at least $1/n$, estimate the support size of $\mathcal{P}$ up to $\pm\varepsilon n$. The state of the art estimator, due to Valiant & Valiant (2011); Wu & Yang (2019), solves this problem using only $O(n/\log n)$ samples (for a constant $\varepsilon$). Both papers also show that this bound is tight.

A more straightforward linear-time algorithm exists, which reports the number of distinct elements seen in a sample of size $N = O(n \log \varepsilon^{-1})$ (which is $O(n)$ for constant $\varepsilon$), without accounting for the unseen items. This algorithm succeeds because each element $i$ with non-zero mass (and thus mass at least $1/n$) appears in the sample with probability at least $1 - (1 - 1/n)^N > 1 - \varepsilon$, so in expectation, at most $\varepsilon \cdot n$ elements with non-zero mass will not appear in the sample. Thus, in general, the number of samples required by the best possible algorithm (i.e., $n/\log n$) is only logarithmically smaller than the complexity of the straightforward linear-time algorithm.

A natural approach to improve over this bound is to leverage the fact that in many applications, the input distribution is not entirely unknown. Indeed, one can often obtain rough approximations of the element frequencies by analyzing different but related distributions. For example, in genomics, frequency estimates can be obtained from the frequencies of genome regions of different species; in linguistics they can be inferred from the statistical properties of the language (e.g., long words are rare), or from a corpus of writings of a different but related author, etc. More generally, such estimates can be learned using modern machine learning techniques, given the true element frequencies in related data sets. The question then becomes whether one can utilize such predictors for use in support size estimation procedures in order to improve the estimation accuracy.

**Our results** In this paper we initiate the study of such "learning-based" methods for support size estimation. Our contributions are both theoretical and empirical. On the theory side, we show that given a "good enough" predictor of the distribution $\mathcal{P}$, one can solve the problem using much fewer than $n/\log n$ samples. Specifically, suppose that in the input distribution $\mathcal{P}$ the probability of element $i$ is $p_i$, and that we have access to a predictor $\Pi(i)$ such that $\Pi(i) \leq p_i \leq b \cdot \Pi(i)$ for some constant approximation factor $b \geq 1$.[3] Then we give an algorithm that estimates the support size up to $\pm\varepsilon n$ using only

$$\log(1/\varepsilon) \cdot n^{1-\Theta(1/\log(1/\varepsilon))}$$

samples, assuming the approximation factor $b$ is a constant (see Theorem 1 for a more detailed bound). This improves over the bound of Wu & Yang (2019) for any fixed values of the accuracy parameter $\varepsilon$ and predictor quality factor $b$. Furthermore, we show that this bound is almost tight.

Our algorithm is presented in Algorithm 1. On a high level, it partitions the range of probability values into geometrically increasing intervals. We then use the predictor to assign the elements observed in the sample to these intervals, and produce a Wu-Yang-like estimate within each interval. Specifically, our estimator is based on Chebyshev polynomials (as in Valiant & Valiant (2011); Wu & Yang (2019)), but the finer partitioning into intervals allows us to use polynomials with different, carefully chosen parameters. This leads to significantly improved sample complexity if the predictor is sufficiently accurate.

On the empirical side, we evaluate the proposed algorithms on a collection of real and synthetic data sets. For the real data sets (network traffic data and AOL query log data) we use neural-network based predictors from Hsu et al. (2019). Although those predictors do not always approximate the true distribution probabilities up to a small factor, our experiments nevertheless demonstrate that the new algorithm offers substantial improvements (up to 3x reduction in relative error) in the estimation accuracy compared to the state of the art algorithm of Wu & Yang (2019).

## 1.1 RELATED WORK

**Estimating support size** As described in the introduction, the problem has been studied extensively in statistics and theoretical computer science. The best known algorithm, due to Wu & Yang

---

[2]This constraint is naturally satisfied e.g., if the distribution $\mathcal{P}$ is an empirical distribution over a data set of $n$ items. In fact, in this case all probabilities are multiples of $1/n$ so the support size is equal to the number of distinct elements in the data set.

[3]Our results hold without change if we modify the assumption to $r \cdot \Pi(i) \leq p_i \leq r \cdot b \cdot \Pi(i)$, for any $r > 0$. We use $r = 1$ for simplicity.

(2019), uses $O(\log^2(1/\varepsilon) \cdot n/\log n)$ samples. Because of the inherent limitations of the model that uses only random samples, Canonne & Rubinfeld (2014) considered an augmented model where an algorithm has access to the *exact* probability of any sampled item. The authors show that this augmentation is very powerful, reducing the sampling complexity to only $O(1/\varepsilon^2)$. More recently, Onak & Sun (2018) proved that Canonne and Rubinfeld's algorithm works as long as the probabilities accessed are accurate up to a $(1 \pm \frac{\varepsilon}{3})$-multiplicative factor. However, this algorithm strongly relies on the probabilities being extremely accurate, and the predicted probabilities even being off by a small constant factor can cause the support size estimate to become massively incorrect. As a result, their algorithm is not robust to mispredicted probabilities, as our experiments show.

A different line of research studied *streaming algorithms* for estimating the number of distinct elements. Such algorithms have access to the whole data set, but must read it in a single pass using limited memory. The best known algorithms for this problem compute a $(1 + \varepsilon)$-approximate estimation to the number of distinct elements using $O(1/\varepsilon^2 + \log n)$ bits of storage (Kane et al., 2010). See the discussion in that paper for a history of the problem and further references.

**Learning-based algorithms**   Over the last few years, there has been a growing interest in using machine learning techniques to improve the performance of "classical" algorithms. This methodology found applications in similarity search (Wang et al., 2016; Sablayrolles et al., 2019; Dong et al., 2020), graph optimization (Khalil et al., 2017; Balcan et al., 2018), data structures (Kraska et al., 2018; Mitzenmacher, 2018), online algorithms (Lykouris & Vassilvitskii, 2018; Purohit et al., 2018), compressed sensing (Mousavi et al., 2015; Baldassarre et al., 2016; Bora et al., 2017) and streaming algorithms (Hsu et al., 2019; Jiang et al., 2019). The last two papers are closest to our work, as they solve various computational problems over data streams, including distinct elements estimation in Jiang et al. (2019) using frequency predictors. Furthermore, in our experiments we are using the neural-network-based predictors developed in Hsu et al. (2019). However, our algorithm operates in a fundamentally different model, using a sublinear (in $n$) number of samples of the input, as opposed to accessing the full input via a linear scan. Thus, our algorithms run in *sublinear time*, in contrast to streaming algorithms that use *sublinear space*.

**Distribution property testing**   This work can be seen more broadly in the context of testing properties of distributions over large discrete domains. Such questions are studied at the crossroads of social networks, statistics, information theory, database algorithms, and machine learning algorithms. Examples of specific properties that have been extensively considered include testing whether the distribution is uniform, Gaussian, high entropy, independent or monotone increasing (see e.g. Rubinfeld (2012); Canonne (2015); Goldreich (2017) for surveys on the topic).

## 2   LEARNING-BASED ALGORITHM

### 2.1   PRELIMINARIES

**Problem setting and notation.**   The support estimation problem is formally defined as follows. We are given sample access to an unknown distribution $\mathcal{P}$ over a discrete domain of size $n$. For simplicity, we identify the domain with $[n] = \{1, \ldots, n\}$. Let $p_i$ denote the probability of element $i$. Let $\mathcal{S}(\mathcal{P}) = \{i : p_i > 0\}$ be the support of $\mathcal{P}$. Our goal is to estimate the support size $S = |\mathcal{S}(\mathcal{P})|$ using as few samples as possible. In particular, given $\varepsilon > 0$, the goal is to output an estimate $\tilde{S}$ that satisfies $\tilde{S} \in [S - \varepsilon n, S + \varepsilon n]$.

We assume that the minimal non-zero mass of any element is at least $1/n$, namely, that $p_i \geq 1/n$ for every $i \in \mathcal{S}(\mathcal{P})$. This is a standard promise in the support estimation problem (see, e.g., Raskhodnikova et al. (2009); Valiant & Valiant (2011); Wu & Yang (2019)), and as mentioned earlier, it naturally holds in the context of counting distinct elements, where $p_i$ is defined as the count of element $i$ in the sample divided by $n$. Furthermore, a lower bound on the minimum non-zero probability is a necessary assumption without which no estimation algorithm is possible, even if the number of samples is allowed to be an arbitrarily large function of $n$, i.e., not just sublinear algorithms. The reason is that there could be arbitrarily many elements with exceedingly small probabilities that would never be observed. See for example the discussion in the supplementary Section 5 of Orlitsky et al. (2016).

In the learning-based setting, we furthermore assume we have a predictor $\Pi$ that can provide information about $p_i$. In our analysis, we will assume that $\Pi(i)$ is a constant factor approximation of each $p_i$. In order to bound the running time of our algorithms, we assume we are given access to a ready-made predictor and (as in Canonne & Rubinfeld (2014)) that evaluating $\Pi(i)$ takes unit time. In our experiments, we use neural network based predictors from Hsu et al. (2019). In general, predictors need to be trained (or otherwise produced) in advance. This happens in a preprocessing stage, before the input distribution is given to the algorithm, and this stage is not accounted for in the sublinear running time. We also note that training the predictor needs to be done only once for all future inputs (not once per input).

**The Wu & Yang (2019) estimator.** In the classical setting (without access to a predictor), Wu & Yang (2019) gave a sample-optimal algorithm based on Chebyshev polynomials. We now describe it briefly, as it forms the basis for our learning-based algorithm.

Suppose we draw $N$ samples, and let $N_i$ be the number of times element $i$ is observed. The output estimate of Wu & Yang (2019) is of the form

$$\tilde{S}_{\text{WY}} = \sum_{i \in [n]} (1 + f(N_i)),$$

where $f(N_i)$ is a correction term intended to compensate for the fact that some elements in the support do not appear in the sample at all. If $p_i = 0$, then necessarily $N_i = 0$ (as $i$ cannot appear in the sample). Thus, choosing $f(0) = -1$ ensures that unsupported elements contribute nothing to $\tilde{S}_{\text{WY}}$. On the other hand, if $p_i > \frac{\log n}{N}$, then by standard concentration we have $N_i > \Omega(\log n)$ with high probability; thus choosing $f(N_i) = 0$ for all $N_i > L = \Omega(\log n)$ ensures that high-mass elements are only counted once in $\tilde{S}_{\text{WY}}$. It remains to take care of elements $i$ with $p_i \in [\frac{1}{n}, \frac{\log n}{N}]$.

By a standard Poissonization trick, the expected additive error $|S - \mathbb{E}[\tilde{S}_{\text{WY}}]|$ can be bounded by $\sum_{i \in [n]} |P_L(p_i)|$, where $P_L$ is the degree-$L$ polynomial

$$P_L(x) = \sum_{k=0}^{L} \frac{\mathbb{E}[N]^k}{k!} \cdot f(k) \cdot x^k.$$

To make the error as small as possible, we would like to choose $f(1), \ldots, f(L)$ so as to minimize $|P_L(p_i)|$ on the interval $p_i \in [\frac{1}{n}, \frac{\log n}{N}]$, under the constraint $P_L(0) = -1$ (which is equivalent to $f(0) = -1$). This is a well-known extremal problem, and its solution is given by Chebyshev polynomials, whose coefficients have a known explicit formula. Indeed, Wu & Yang (2019) show that choosing $f(1), \ldots, f(L)$ such that $\frac{\mathbb{E}[N]^k}{k!} f(k)$ are the coefficients of an (appropriately shifted and scaled) Chebyshev polynomial leads to an optimal sample complexity of $O(\log^2(1/\varepsilon) \cdot n/\log n)$.

## 2.2 OUR ALGORITHM

Our main result is a sample-optimal algorithm for estimating the support size of an unknown distribution, where we are given access to samples as well as approximations to the probabilities of the elements we sample. Our algorithm is presented in Algorithm 1. It partitions the interval $[\frac{1}{n}, \frac{\log n}{N}]$ into geometrically increasing intervals $\{[\frac{b^j}{n}, \frac{b^{j+1}}{n}] : j = 0, 1, \ldots\}$, where $b$ is a fixed constant that we refer to as the *base* parameter (in our proofs this parameter upper bounds the approximation factor of the predictor, which is why we use the same letter to denote both; its setting in practice is studied in detail in the next section). The predictor assigns the elements observed in the sample to intervals, and the algorithm computes a Chebyshev polynomial estimate within each interval. Since the approximation quality of Chebyshev polynomials is governed by the ratio between the interval endpoints (as well as the polynomial degree), this assignment can be leveraged to get more accurate estimates, improving the overall sample complexity. Our main theoretical result is the following.

**Theorem 1.** *Let $b > 1$ be a fixed constant. Suppose we have a predictor that given $i \in [n]$ sampled from the input distribution, outputs $\Pi(i)$ such that $\Pi(i) \leq p_i \leq b \cdot \Pi(i)$. Then, for any $\varepsilon > n^{-1/2}$, if we set $L = O(\log(1/\varepsilon))$ and draw $N \sim Poisson\left(L \cdot n^{1-1/L}\right)$ samples, Algorithm 1 reports an estimate $\tilde{S}$ that satisfies $\tilde{S} \in [S - \varepsilon\sqrt{nS}, S + \varepsilon\sqrt{nS}]$ with probability at least $9/10$. In other words,*

---

**Algorithm 1:** Learning-Based Support Estimation

---

**Input:** Number of samples $N$, domain size $n$, base $b$, polynomial degree $L$, predictor $\Pi$

**Output:** Estimate $\tilde{S}$ of the support size

1 Partition $[\frac{1}{n}, 1]$ into intervals $I_j = \left[\frac{b^j}{n}, \frac{b^{j+1}}{n}\right]$ for $j = 0, \ldots, \log_b n$

2 Let $a_0, a_1, \ldots, a_L$ be the coefficient of the Chebyshev polynomial $P_L(x)$ from Equation (1)
   (Note that $a_k = 0$ for all $k > L$)

3 Draw $N$ random samples

4 $N_i = \#$ of times we see element $i$ in samples

5 **for** *every interval $I_j$* **do**

6     **if** $\frac{b^j}{n} \leq \frac{0.5 \log n}{N}$ **then**

7        $\tilde{S}_j = \sum_{i \in [n] : \Pi(i) \in I_j} \left(1 + a_{N_i} \left(\frac{n}{b^j}\right)^{N_i} \cdot \frac{N_i!}{N^{N_i}}\right)$

8     **else**

9        $\tilde{S}_j = \#\{i \in [n] : N_i \geq 1, \Pi(i) \in I_j\}$

10     **end**

11 **end**

12 **return** $\tilde{S} = \sum_{j=0}^{\log_b n} \tilde{S}_j$

---

*using*

$$O\left(\log(1/\varepsilon) \cdot n^{1 - \Theta(1/\log(1/\varepsilon))}\right)$$

*samples, we can approximate the support size $S$ up to an additive error of $\varepsilon\sqrt{nS}$, with probability at least $9/10$.*

Note that sample complexity of Wu & Yang (2019) (which is optimal without access to a predictor) is nearly linear in $n$, while Theorem 1 gives a bound that is polynomially smaller than $n$ for every fixed $\varepsilon > 0$. Also, note that $\sqrt{nS} \leq n$, so our estimate $\tilde{S}$ is also within $\varepsilon \cdot n$ of the support size $S$.

We also prove a corresponding lower bound, proving that the above theorem is essentially tight.

**Theorem 2.** *Suppose we have access to a predictor that returns $\Pi(i)$ such that $\Pi(i) \leq p_i \leq 2 \cdot \Pi(i)$ for all sampled $i$. Then any algorithm that estimates the support size up to an $\varepsilon n$ additive error with probability at least $9/10$ needs $\Omega(n^{1 - \Theta(1/\log(1/\varepsilon))})$ samples.*

We prove Theorems 1 and 2 in Appendix A. We note that while our upper bound proof follows a similar approach to Wu & Yang (2019), our lower bound follows a combinatorial approach differing from their linear programming arguments.

**The Chebyshev polynomial.** For completeness, we explicitly write the polynomial coefficients used by Algorithm 1. The standard Chebyshev polynomial of degree $L$ on the interval $[-1, 1]$ is given by[4]

$$Q_L(x) = \cos(L \cdot \arccos(x)).$$

For Algorithm 1, we want a polynomial as follows:

$$P_L(x) = \sum_{k=0}^{L} a_k x^k \text{ satisfying } P_L(0) = -1 \text{ and } P_L(x) \in [-\varepsilon, \varepsilon] \text{ for all } 1 \leq x \leq b^2.$$

This is achieved by shifting and scaling $Q_L$, namely, this polynomial can be written as

$$P_L(x) = -\frac{Q_L\left(\frac{2x - (b^2+1)}{(b^2-1)}\right)}{Q_L\left(-\frac{b^2+1}{b^2-1}\right)}, \tag{1}$$

---

[4]The fact this is indeed a polynomial is proven in standard textbooks, e.g., Timan et al. (1963).

where $\varepsilon$ equals $\left| Q_L \left( -\frac{b^2+1}{b^2-1} \right) \right|^{-1}$, which decays as $e^{-\Theta(L)}$ if $b$ is a constant. Thus it suffices to choose $L = O(\log(1/\varepsilon))$ in Theorem 1.

## 2.3 ASSUMPTIONS ON THE PREDICTOR'S ACCURACY

In Theorems 1 and 2, we assume that the predictor $\Pi$ is always correct within a constant factor for each element $i$. As we will discuss in the experimental section, the predictor that we use does not always satisfy this property. Hence, perhaps other models of the accuracy of $\Pi$ are better suited, such as a promised upper bound on $TV(\Pi, \mathcal{P})$, i.e., the total variation distance between $\Pi$ and $\mathcal{P}$. However, it turns out that if we are promised that $TV(\Pi, \mathcal{P}) \leq \varepsilon$, then the algorithm of Canonne & Rubinfeld (2014) can in fact approximate the support size of $\mathcal{P}$ up to error $O(\varepsilon) \cdot n$ using only $O(\varepsilon^{-2})$ samples. Conversely, if we are only promised that $TV(\Pi, \mathcal{P}) \leq \gamma$ for some $\gamma \gg \varepsilon$, the algorithm of Wu & Yang (2019), which requires $\Theta(n/\log n)$ samples to approximate the support size of $\mathcal{P}$, is optimal. We formally state and prove both of these results in Appendix A.3.

Our experimental results demonstrate that our algorithm can perform better than both the algorithms of Canonne & Rubinfeld (2014) and Wu & Yang (2019). This suggests that our assumption (a point-wise approximation guarantee, but with an arbitrary constant multiplicative approximation factor), even if not fully accurate, is better suited for our application than a total variation distance bound.

## 3 EXPERIMENTS

In this section we evaluate our algorithm empirically on real and synthetic data.

**Datasets.** We use two real and one synthetic datasets:

- **AOL**: 21 million search queries from 650 thousand users over 90 days. The queries are keywords for the AOL search engine. Each day is treated as a separate input distribution. The goal is to estimate the number of distinct keywords.

- **IP**: Packets collected at a backbone link of a Tier1 ISP between Chicago and Seattle in 2016 over 60 minutes.[5] Each packet is annotated with the sender IP address, and the goal is to estimate the number of distinct addresses. Each minute is treated as a separate distribution.

- **Zipfian**: Synthetic dataset of samples drawn from a finite Zipfian distribution over $\{1, \ldots, 10^5\}$ with the probability of each element $i$ proportional to $i^{-0.5}$.

The AOL and IP datasets were used in Hsu et al. (2019), who trained a recurrent neural network (RNN) to predict the frequency of a given element for each of those datasets. We use their trained RNNs as predictors. For the Zipfian dataset we use the empirical counts of an independent sample as predictions. A more detailed account of each predictor is given later in this section. The properties of the datasets are summarized in Table 1.

**Baselines.** We compare Algorithm 1 with two existing baselines:

- The algorithm of Wu & Yang (2019), which is the state of the art for algorithms without predictor access. We abbreviate its name as WY.[6]

- The algorithm of Canonne & Rubinfeld (2014), which is the state of the art for algorithms with access to a *perfect* predictor. We abbreviate its name as CR.

**Error measurement.** We measure accuracy in terms of the relative error $|1 - \tilde{S}/S|$, where $S$ is the true support size and $\tilde{S}$ is the estimate returned by the algorithm. We report median errors over $50$ independent executions of each experiment, $\pm$ one standard deviation.

---

[5]From CAIDA internet traces 2016, `https://www.caida.org/data/monitors/passive-equinix-chicago.xml`.

[6]Apart from their tight analysis, Wu & Yang (2019) also report experimental results showing their algorithm is empirically superior to previous baselines.

Table 1: Datasets used in our experiments. The listed values of $n$ (total size) and support size (distinct elements) for AOL/IP are per day/minute (respectively), approximated across all days/minutes.

| Name | Type | # Distributions | Predictor | $n$ | Support size |
|---|---|---|---|---|---|
| AOL | Keywords | 90 (days) | RNN | $\sim 4 \cdot 10^5$ | $\sim 2 \cdot 10^5$ |
| IP | IP addresses | 60 (minutes) | RNN | $\sim 3 \cdot 10^7$ | $\sim 10^6$ |
| Zipfian | Synthetic | 1 | Empirical | $\sim 2 \cdot 10^5$ | $10^5$ |

**Summary of results.**   Our experiments show that on one hand, our algorithm can indeed leverage the predictor to get significantly improved accuracy compared to WY. On the other hand, our algorithm is robust to different predictors: while the CR algorithm performs extremely well on one dataset (AOL), it performs poorly on the other two (IP and Zipfian), whereas our algorithm is able to leverage the predictors in those cases too and obtain significant improvement over both baselines.

## 3.1   BASE PARAMETER SELECTION

Algorithm 1 uses two parameters that need to be set: The polynomial degree $L$, and the base parameter $b$. The performance of the algorithm is not very sensitive to $L$, and for simplicity we use the same setting as Wu & Yang (2019), $L = \lfloor 0.45 \log n \rfloor$. The setting of $b$ requires more care.

Recall that $b$ is a constant used as the ratio between the maximum and minimum endpoint of each interval $I_j$ in Algorithm 1. There are two reasons why $b$ cannot be chosen too small. One is that the algorithm is designed to accommodate a predictor that provides a $b$-approximation of the true probabilities, so larger $b$ makes the algorithm more robust to imperfect predictors. The other reason is that small $b$ means using many small intervals, and thus a smaller number of samples assigned to each interval. This leads to higher noise in the Chebyshev polynomial estimators invoked within each interval (even if the assignment of elements to intervals is correct), and empirically impedes performance. On the other hand, if we set $b$ to be too large (resulting in one large interval the covers almost the whole range of $p_i$'s), we are essentially not using information from the predictor, and thus do not expect to improve over WY.

To resolve this issue, we introduce a sanity check in each interval $I_j$, whose goal is to rule out bases that are too small. The sanity check passes if $\tilde{S}_j \in [0, 1/l_j]$, where $l_j$ is the left endpoint of $I_j$ (i.e., $I_j = [l_j, r_j]$), and $\tilde{S}_j$ is as defined in Algorithm 1. The reasoning is as follows. On one hand, the Chebyshev polynomial estimator which we use to compute $\tilde{S}_j$ can in fact return negative numbers, leading to failure modes with $\tilde{S}_j < 0$. On the other hand, since all element probabilities in $I_j$ are lower bounded by $l_j$, it can contain at most $1/l_j$ elements. Therefore, any estimate $\tilde{S}_j$ of the number of elements in $I_j$ which is outside $[0, 1/l_j]$ is obviously incorrect.

In our implementation, we start by running Algorithm 1 with $b = 2$. If any interval fails the sanity check, we increment $b$ and repeat the algorithm (with the same set of samples). The final base we use is twice the minimal one such that all intervals pass the sanity check, where the final doubling is to ensure we are indeed past the point where all checks succeed.

The effect of this base selection procedure on each of our datasets is depicted in Figures 1, 3, and 5.[7] For a fixed sample size, we plot the performance of our algorithm (dotted blue) as the base increases compared to WY (solid orange, independent of base), as well as the fraction of intervals that failed the sanity check (dashed green). The plots show that for very small bases, the sanity check fails on some intervals, and the error is large. When the base is sufficiently large, all intervals pass the sanity check, and we see a sudden plunge in the error. Then, as the base continues to grow, our algorithm continues to perform well, but gradually degrades and converges to WY due to having a single dominating interval. This affirms the reasoning above and justifies our base selection procedure.

---

[7]In those plots, for visualization, in order to compute the error whenever a sanity check fails in $I_j$, we replace $\tilde{S}_j$ with a naïve estimate, defined as the number of distinct sample elements assigned to $I_j$. Note that our implementation never actually uses those naïve estimates, since it only uses bases that pass all sanity checks.

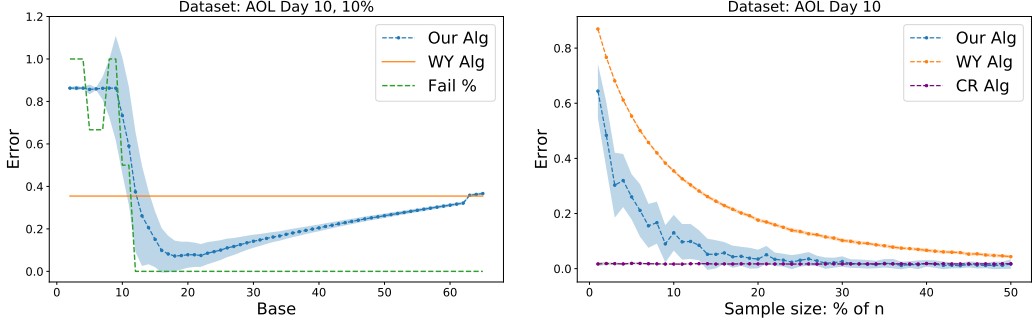

Figure 1: Error by base, AOL, sample size $10\% \cdot n$

Figure 2: Error per sample size, AOL

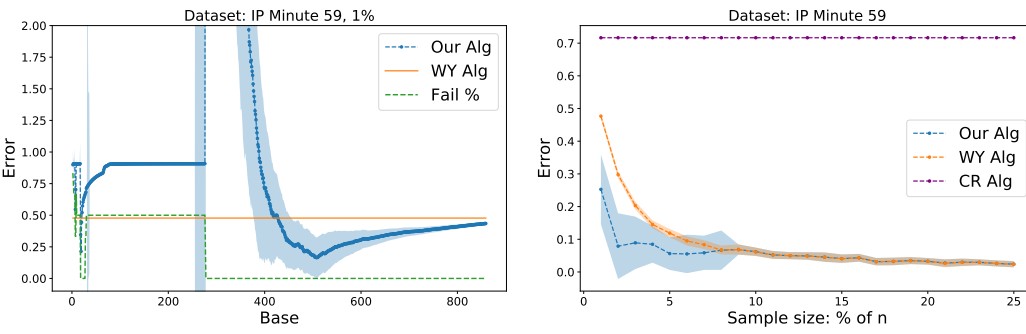

Figure 3: Error by base, IP, sample size $1\% \cdot n$

Figure 4: Error per sample size, IP

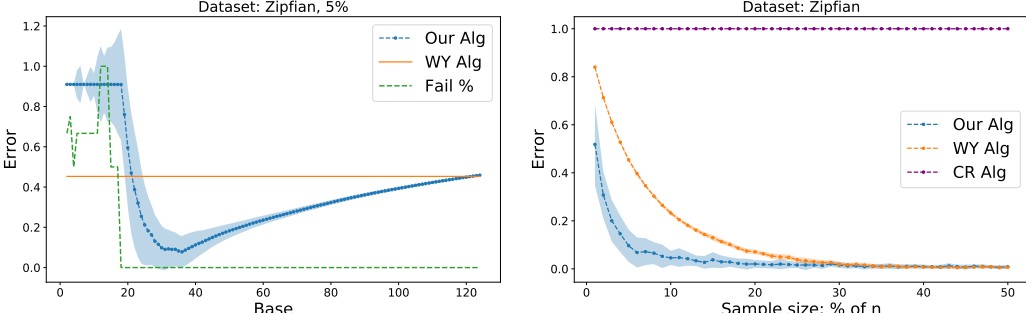

Figure 5: Error by base, Zipfian, sample size $5\%n$

Figure 6: Error per sample size, Zipfian

## 3.2 RESULTS

**AOL data.** As the predictor, we use the RNN trained by Hsu et al. (2019) to predict the frequency of a given keyword. Their predictor is trained on the first 5 days and the 6th day is used for validation. The results for day #10 are shown in Figure 2. (The results across all days are similar; see more below.) They show that our algorithm performs significantly better than WY. Nonetheless, CR (which relies on access to a perfect predictor) achieves better performance on a much smaller sample size. This is apparently due to highly specific traits of the predictor; as we will see presently, the performance of CR is considerably degraded on the other datasets.

**IP data.** Here too we use the trained RNN of Hsu et al. (2019). It is trained on the first 7 minutes and the 8th minute is used for validation. However, unlike AOL, Hsu et al. (2019) trained the RNN to predict the *log* of the frequency of the given IP address, rather than the frequency itself, due to training stability considerations. To use it as a predictor, we exponentiate the RNN output. This inevitably leads to less accurate predictions.

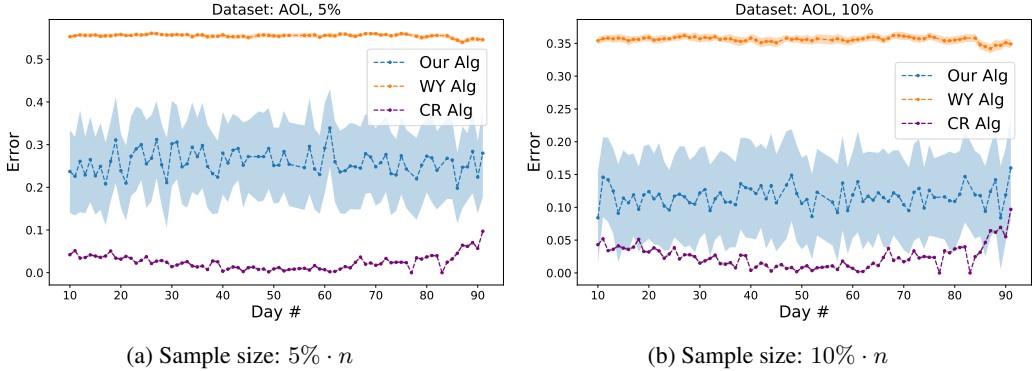

(a) Sample size: $5\% \cdot n$

(b) Sample size: $10\% \cdot n$

Figure 7: Error across the different AOL days

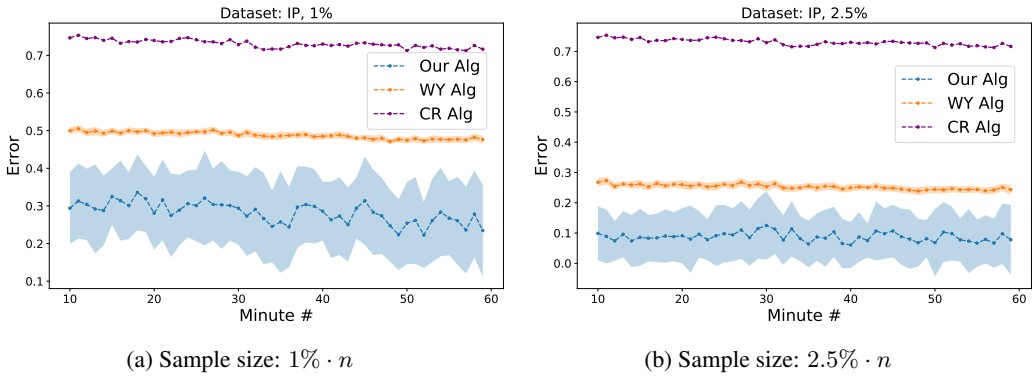

(a) Sample size: $1\% \cdot n$

(b) Sample size: $2.5\% \cdot n$

Figure 8: Error across the different IP minutes

The results for minute 59 are shown in. Figure 4. (As in the AOL data, the results across all minutes are similar; see more below). Again we see a significant advantage to our algorithm over WY for small sample sizes. Here, unlike the AOL dataset, CR does not produce good results.

**Zipfian data.** To form a predictor for this synthetic distribution, we drew a random sample of size $10\%$ of $n$, and used the empirical count of each element in this fixed sample as the prediction for its frequency. If the predictor is queried for an element that did not appear in the sample, its predicted probability is reported as the minimum $1/n$. We use this fixed predictor in all repetitions of the experiment (which were run on fresh independent samples). The results are reported in Figure 6. As with the IP data, our algorithm significantly improves over WY for small sample sizes, and both algorithms outperform CR by a large margin.

**AOL and IP results over time.** Finally, we present accuracy results over the days/minutes of the AOL/IP datasets (respectively). The purpose is to demonstrate that the performance of our algorithm remains consistent over time, even when the data has moved away from the initial training period and the predictor may become 'stale'. The results for AOL are shown in Figures 7a, 7b, yielding a median **2.2-fold** and **3.0-fold** improvement over WY (for sample sizes $5\% \cdot n$ and $10\% \cdot n$, respectively). The results for IP are shown in Figures 8a and 8b, yielding a median **1.7-fold** and **3.0-fold** improvement over WY (for sample sizes $1\% \cdot n$ and $2.5\% \cdot n$, respectively). As before, CR performs better than either algorithm on AOL, but fails by a large margin on IP.

ACKNOWLEDGMENTS

This research was supported in part by the NSF TRIPODS program (awards CCF-1740751 and DMS-2022448); NSF awards CCF-1535851, CCF-2006664 and IIS-1741137; Fintech@CSAIL; Simons Investigator Award; MIT-IBM Watson collaboration; Eric and Wendy Schmidt Fund for

Strategic Innovation, Ben-Gurion University of the Negev; MIT Akamai Fellowship; and NSF Graduate Research Fellowship Program.

The authors would like to thank Justin Chen for helpful comments on a draft of this paper, as well as the anonymous reviewers.

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

## A    OMITTED PROOFS

### A.1    ANALYSIS OF THE UPPER BOUND

In this subsection, we prove Theorem 1, which provides the upper bound for the sample complexity of Algorithm 1.

*Proof of Theorem 1.* Consider some $i \in [n]$. The predictor $\Pi$ gives us an estimate $\Pi(i) \in [p_i, b \cdot p_i]$ of the unknown true probability $p_i$. Let $j_i$ be the integer for which $\Pi(i) \in [\frac{b^{j_i}}{n}, \frac{b^{j_i+1}}{n}]$. We treat $i$ as assigned by the predictor to the interval $I_{j_i}$. Observe that as a consequence, we have

$$p_i \in \left[\frac{b^{j_i}}{n}, \frac{b^{j_i+2}}{n}\right]. \tag{2}$$

Suppose $i$ is observed $k$ times in the sample. Define,

$$\tilde{S}(i) = \begin{cases} 1 + a_k \left(\frac{n}{b^j}\right)^k \cdot \frac{k!}{N^k} & \text{if } \frac{b^{j_i}}{n} \le \frac{0.5 \log n}{N} \\ 1 & \text{if } \frac{b^{j_i}}{n} > \frac{0.5 \log n}{N} \text{ and } k \ge 1 \\ 0 & \text{if } \frac{b^{j_i}}{n} > \frac{0.5 \log n}{N} \text{ and } k = 0 \end{cases}$$

The $a_k$'s are the Chebyshev polynomial coefficients as per Algorithm 1. Note that as we drew a total of $Pois(N)$ samples, the number of times each element $i$ appears in the sample is distributed as $N_i = Pois(N \cdot p_i)$ times, and the values $N_i$ are independent across different $i$'s (this is the standard Poissonization trick). Thus the $\tilde{S}(i)$'s are also independent. Moreover, note that the value $\tilde{S}_j$ defined in Algorithm 1 satisfies $\tilde{S}_j = \sum_{i:j_i=j} \tilde{S}(i)$. Finally, by eq. (2) we have $p_i \cdot \frac{n}{b^{j_i}} \in [1, b^2]$. Thus, if $\frac{b^{j_i}}{n} \le \frac{0.5 \log n}{N}$, we compute the expectation of $\tilde{S}(i)$ as (letting $j = j_i$ to ease notation),

$$\begin{aligned}
\mathbb{E}[\tilde{S}(i)] &= 1 + \sum_{k=0}^{L} \mathbb{P}(N_i = k) \cdot a_k \left(\frac{n}{b^j}\right)^k \cdot \frac{k!}{N^k} \\
&= 1 + \sum_{k=0}^{L} \frac{e^{-Np_i}(Np_i)^k}{k!} \cdot a_k \left(\frac{n}{b^j}\right)^k \cdot \frac{k!}{N^k} \\
&= 1 + e^{-Np_i} \cdot \sum_{k=0}^{L} a_k \left(p_i \cdot \frac{n}{b^j}\right)^k \\
&= 1 + e^{-Np_i} \cdot P_L \left(p_i \cdot \frac{n}{b^j}\right) \\
&\in [1 - \varepsilon, 1 + \varepsilon],
\end{aligned}$$

since $p_i \cdot \frac{n}{b^j} \in [1, b^2]$ and since $e^{-Np_i} \le 1$. However, if $\frac{b^{j_i}}{n} > \frac{0.5 \log n}{N}$, then $\mathbb{E}[\tilde{S}(i)]$ equals the probability that $i$ is sampled, which is $1 - (1 - p_i)^N$. But since $p_i > \frac{b^{j_i}}{n} > \frac{0.5 \log n}{N}$, we have that $1 \ge \mathbb{E}[\tilde{S}(i)] > 1 - \left(1 - \frac{0.5 \log n}{N}\right)^N > 1 - e^{(0.5 \log n/N) \cdot N} = 1 - n^{-1/2}$. We note that if $i$ is never seen (which is possible even if $p_i \ne 0$), we do not know $j_i$. However, in this case, $\tilde{S}(i) = 0$ regardless of the value of $j_i$. Therefore, we get an estimator $\tilde{S}(i)$ such that $\tilde{S}(i) = 0$ if $p_i = 0$ and $\mathbb{E}[\tilde{S}(i)] \in [1 - \varepsilon, 1 + \varepsilon]$ otherwise, assuming $\varepsilon > n^{-1/2}$.

Next we analyze the variance of $\tilde{S}(i)$. For $i$ with $p_i = 0$, $\tilde{S}(i) = 0$ always, so $\mathbf{Var}(\tilde{S}(i)) = 0$. Otherwise, if $\frac{b^{j_i}}{n} > \frac{0.5 \log n}{N}$, then $\tilde{S}(i)$ is always between 0 and 1, so $\mathbf{Var}(\tilde{S}(i)) \le 1$. Finally, if

$\frac{b^{j_i}}{n} \leq \frac{0.5 \log n}{N}$, then $\mathbf{Var}(\tilde{S}(i)) \leq \mathbb{E}[(\tilde{S}(i) - 1)^2]$ and we can write (again with $j = j_i$)

$$\mathbb{E}[(\tilde{S}(i) - 1)^2] = \sum_{k=0}^{L} \mathbb{P}(N_i = k) \cdot \left( a_k \left( p_i \cdot \frac{n}{b^j} \right)^k \right)^2$$

$$= \sum_{k=0}^{L} \frac{e^{-Np_i}(Np_i)^k}{k!} \cdot \left( a_k \left( \frac{n}{b^j} \right)^k \cdot \frac{k!}{N^k} \right)^2$$

$$\leq \left( \max_{0 \leq k \leq L} a_k^2 \right) \cdot \sum_{k=0}^{L} \left( p_i \cdot \frac{n}{b^j} \right)^k \cdot \left( \frac{n}{b^j} \right)^k \cdot \frac{k!}{N^k}$$

$$\leq \left( \max_{0 \leq k \leq L} a_k^2 \right) \cdot \sum_{k=0}^{L} \left( \frac{b^2 \cdot k \cdot n}{N} \right)^k, \tag{3}$$

where for the last inequality we noted that $k! \leq k^k$, $p_i \cdot \frac{n}{b^j} \leq b^2$, and $b^j \geq 1$.

Now, it is a well known consequence of the Markov Brothers' inequality that all the coefficients of the standard degree $L$ Chebyshev polynomial $Q_L(x)$ are bounded by $e^{O(L)}$ (Markov, 1892). Since $P_L = \sum_{k=0}^{L} a_k x^k$ is just $\varepsilon \cdot Q \left( \frac{b^2+1}{2} + x \cdot \frac{b^2-1}{2} \right)$, we have that for any fixed $b = O(1)$, the coefficients $a_k$ are all bounded by $e^{O(L)}$ as well. Therefore, for $N = C \cdot b^2 \cdot L \cdot n^{1-1/L}$ for some constant $C$, we have that $b^2 \cdot k \cdot n / N \leq n^{1/L}/C$, so we can bound Equation (3) by

$$e^{O(L)} \cdot \sum_{k=0}^{L} \left( \frac{n^{1/L}}{C} \right)^k \leq \frac{n}{(C')^L}$$

for some other constant $C'$.

In summary, if $p_i \neq 0$, we have that $\mathbb{E}[\tilde{S}(i)] \in [1 - \varepsilon, 1 + \varepsilon]$ and $\mathbf{Var}(\tilde{S}(i)) \leq \varepsilon^2 n$ if we choose $C$ to be a sufficiently large constant, since $\varepsilon = e^{-\Theta(L)}$ for a constant $1 < b \leq O(1)$. This is true even for $\frac{b^{j_i}}{n} > \frac{0.5 \log n}{N}$ since $\varepsilon^2 n > 1$. However, we know that $\tilde{S} = \sum_{j=0}^{\log_b n} \tilde{S}_j = \sum_{i=1}^{n} \tilde{S}(i)$, since by the definition of $\tilde{S}_j$, $\tilde{S}_j = \sum_{i:j_i=j} \tilde{S}(i)$. Therefore, since the estimators $\tilde{S}(i)$ are independent, we have that $\mathbb{E}[\tilde{S}] = \sum_{i=1}^{n} \mathbb{E}[\tilde{S}(i)] \in (1 - \varepsilon, 1 + \varepsilon) \cdot S$ and $\mathbf{Var}(\tilde{S}) = \sum_{i:p_i \neq 0} \mathbf{Var}(\tilde{S}(i)) \leq \varepsilon^2 \cdot n \cdot S$, where we use the independence of the $\tilde{S}(i)$'s. Therefore, since $S \leq n$, with probability at least $0.9$, $|\tilde{S} - S| = O(\varepsilon) \cdot \sqrt{n \cdot S}$ by Chebyshev's inequality. $\qquad \square$

**Remark.** *We note that our analysis actually works (and becomes somewhat simpler) even if the algorithm is modified to define $\tilde{S}_j = \sum_{i \in [n]:\Pi(i) \in I_j} \left( 1 + a_{N_i} \left( \frac{n}{b^j} \right)^{N_i} \cdot \frac{N_i!}{N^{N_i}} \right)$ for all intervals, as opposed to $\tilde{S}_j = \#\{i \in [n] : N_i \geq 1, \Pi(i) \in I_j\}$ for the intervals $I_j$ with $\frac{b^j}{n} > \frac{0.5 \log n}{N}$. However, because we can decrease the bias as well as the variance for these larger intervals, we modify the algorithm accordingly. While this does not affect the theoretical guarantees, it demonstrated an improvement in practice. This threshold was also used in Wu & Yang (2019) (the choice of leading constant $0.5$ in the threshold term $\frac{0.5 \log n}{N}$ is arbitrary, and for simplicity we have adopted the value they used).*

## A.2 Analysis of the Lower Bound

In this subsection, we prove Theorem 2, which proves that the sample complexity of Algorithm 1 is essentially tight.

*Proof of Theorem 2.* In order to prove the lower bound, we shall define two distributions $P$ and $Q$ for which the first $k$ moments are matching, but their support size differs on at least $\varepsilon n$ elements. Furthermore, both distributions will be supported on $\{0\} \cup \left[ \frac{k}{n}, \frac{k+1}{n}, \dots, \frac{2k}{n} \right]$, so that a 2-factor approximation predictor does not provide any useful information to an algorithm trying to distinguish the two distributions.

We start with an overview of the definition of the two distributions and then explain how to formalize the arguments, following the Poissonnization and rounding techniques detailed in Raskhodnikova et al. (2009).

Let $\varepsilon = \left( k \cdot 2^{k-1} \cdot \binom{2k}{k} \right)^{-1}$ for some integer $k \geq 1$. Note that $\varepsilon = e^{-\Theta(k)}$. In order to define the distributions, we define $k+1$ real numbers $a_0, \ldots, a_k$ as $a_i = \frac{(-1)^i \cdot \binom{k}{i}}{2^{k-1} \cdot (k+i)}$ for every $i \in \{0, \ldots, k\}$. Suppose for now that $n$ is a multiple of the least common multiple of $2^{k-1}, k, \ldots, 2k$, so that $a_i \cdot n$ is an integer for every $i$. We define $P$ and $Q$ as follows:

- **The distribution** $P$**:** For every $a_i$ such that $a_i > 0$, the support of $P$ contains $a_i \cdot n$ elements $j$ that have probability $p_j = a_i \cdot \frac{k+i}{n}$ each.

- **The distribution** $Q$**:** For every $a_i$ such that $a_i < 0$, the support of $Q$ contains $-a_i \cdot n$ elements $j$ that have probability $q_j = -a_i \cdot \frac{k+i}{n}$ each.

First we prove that $P$ and $Q$ are valid distributions and that all of their non-zero probabilities are greater than $1/n$.

**Claim A.1.** *$P$ and $Q$ as defined above are distributions. Furthermore, their probability values are either $0$ or greater than $1/n$.*

*Proof.* The second part of the claim follows directly from the definition of $P$ and $Q$. We continue to prove the first part, that $P$ and $Q$ are indeed distributions. It holds for $P$ that

$$\sum_{a_i \mid a_i > 0} (na_i) \cdot \frac{k+i}{n} = \sum_{\substack{i=0 \\ i \text{ even}}}^{k} n \cdot \frac{1}{k+i} \cdot \binom{k}{i} \cdot \frac{1}{2^{k-1}} \cdot \frac{k+i}{n} = \frac{1}{2^{k-1}} \cdot \sum_{\substack{i=0 \\ i \text{ even}}}^{k} \binom{k}{i} = 1.$$

Similarly, for $Q$,

$$\sum_{a_i < 0} (n(-a_i)) \cdot \frac{k+i}{n} = \sum_{\substack{i=0 \\ i \text{ odd}}}^{k} n \cdot \frac{1}{k+i} \cdot \binom{k}{i} \cdot \frac{1}{2^{k-1}} \cdot \frac{k+i}{n} = \frac{1}{2^{k-1}} \cdot \sum_{\substack{i=0 \\ i \text{ odd}}}^{k} \binom{k}{i} = 1. \qquad \square$$

We continue to prove that the first $k$ moments of $P$ and $Q$ are matching, and that their support size differs by $\varepsilon n$.

**Claim A.2.** *Let $a_1, \ldots, a_k$ be defined as above. Then*

- *For any $r \in \{1, \ldots, k\}$, it holds that $\sum_{i=0}^{k} a_i \cdot (k+i)^r = 0$.*

- *$\sum_{i=0}^{k} a_i = \varepsilon$.*

*Proof.* By plugging the $a_i$'s as defined above,

$$\sum_{i=0}^{k} a_i \cdot (k+i)^r = \sum_{i=0}^{k} \frac{(-1)^i \binom{k}{i}}{2^{k-1} \cdot (k+i)} \cdot (k+i)^r = \frac{1}{2^{k-1}} \sum_{i=0}^{k} (-1)^i \binom{k}{i} \cdot (k+i)^{r-1}.$$

Hence, letting $r' = r - 1$, it suffices to prove that $\sum_{i=0}^{k} (-1)^i \binom{k}{i} (k+i)^{r'} = 0$ for all $0 \leq r' \leq k-1$. For any fixed $k$, note that since $(k+i)^{r'}$ is a degree $r'$ polynomial in $i$, $(k+i)^{r'}$ can be written as a linear combination of $\binom{i}{s}$ for $0 \leq s \leq r'$ with coefficients $b_0, \ldots, b_{r'}$. Therefore, we would like to prove that: $\sum_{i=0}^{k} (-1)^i \binom{k}{i} \sum_{s=0}^{r'} b_s \binom{i}{s} = 0$. Fix some $s$ in $\{0, \ldots, r'\}$: it suffices to show that for any integer $k$, $\sum_{i=0}^{k} (-1)^i \binom{k}{i} \binom{i}{s} = 0$ for all $0 \leq s \leq k-1$.

Since $\binom{k}{i} \cdot \binom{i}{s} = \binom{k}{k-i, i-s, s} = \binom{k}{s} \cdot \binom{k-s}{i-s}$, by setting $k' = k - s$ and $i' = i - s$, we get

$$\sum_{i=0}^{k}(-1)^i \binom{k}{i}\binom{i}{s} = \sum_{i=0}^{k}(-1)^i \binom{k}{s} \cdot \binom{k-s}{i-s} = \binom{k}{s} \cdot \sum_{i'=0}^{k'}(-1)^{i'+s} \binom{k'}{i'}$$

$$= \binom{k}{s} \cdot (-1)^s \cdot (1-1)^{k'} = 0,$$

where the last equality is since $k' = k - s \geq 1$. This concludes the proof of the first item.

We continue to prove the second item in the claim:

$$\sum_{i=0}^{k} a_i = \varepsilon. \tag{4}$$

Recall that $a_i = \frac{(-1)^i \cdot \binom{k}{i}}{2^{k-1} \cdot (k+i)}$, and that $\varepsilon = \left( k \cdot 2^{k-1} \cdot \binom{2k}{k} \right)^{-1}$, so plugging these into Equation (4) and multiplying both sides by $2^{k-1} \cdot \binom{2k}{k}$, this is equivalent to proving

$$\frac{1}{k} = \sum_{i=0}^{k} \frac{(-1)^i}{k+i} \cdot \binom{k}{i}\binom{2k}{k}. \tag{5}$$

Since $\binom{k}{i} \cdot \binom{2k}{k} = \binom{2k}{k, i, k-i} = \binom{2k}{k+i} \cdot \binom{k+i}{k}$, the right-hand side of Equation (5) equals

$$\sum_{i=0}^{k} \frac{(-1)^i}{k+i} \cdot \binom{2k}{k+i}\binom{k+i}{k} = \sum_{i=0}^{k}(-1)^i \cdot \binom{2k}{k+i} \cdot \binom{k+i-1}{k-1} \cdot \frac{1}{k}.$$

Multiplying by $k$, it suffices to show that

$$1 = \sum_{i=0}^{k}(-1)^i \cdot \binom{2k}{k+i} \cdot \binom{k+i-1}{k-1}. \tag{6}$$

To do this, let $j = k + i$. Then, note that $\binom{k+i-1}{k-1} = \binom{j-1}{k-1} = \frac{(j-1)\cdots(j-k+1)}{(k-1)!}$ which is a degree $k - 1$ polynomial in $j$. For $1 \leq j \leq k - 1$ the polynomial equals 0, but for $j = 0$ the polynomial equals $(-1)^{k-1}$. Therefore, the right hand side of Equation (6) equals

$$\sum_{j=1}^{2k}(-1)^{j-k} \cdot \binom{2k}{j} \cdot \frac{(j-1)\cdots(j-k+1)}{(k-1)!} = 1 + (-1)^{-k} \cdot \sum_{j=0}^{2k}(-1)^j \cdot \binom{2k}{j} \cdot \frac{(j-1)\cdots(j-k+1)}{(k-1)!}.$$

As proven in the first part of this proof, for any $P(j)$ of degree $0 \leq r \leq 2k - 1$, $\sum_{j=0}^{2k}(-1)^j \binom{2k}{j} P(j) = 0$. Therefore, the summation on the right hand side is 0, so this simplifies to 1, as required. □

The following corollary follows directly from the definition of $P$ and $Q$ and the previous claim.

**Corollary A.1.** *The following two items hold for $P$ and $Q$ as defined above.*

- $\mathbb{E}[P^r] = \mathbb{E}[Q^r]$ *for all* $r \in \{1, \ldots, k\}$.

- $TV(P, Q) = \varepsilon n$.

The above corollary states that indeed $P$ and $Q$ as defined above have matching moments for $r = 1$ to $k$, and that they differ by $\varepsilon n$ in their support size. This concludes the high level view of the construction of the distributions $P$ and $Q$. In order to finalize the proof we rely on the standard Possionization and rounding techniques.

First, by Theorem 5.3 in Raskhodnikova et al. (2009), any $s$-samples algorithm can be simulated by an $O(s)$-Poisson algorithm. Hence, we can assume that the algorithm takes $Poi(s)$ samples, rather than an arbitrary number $s$. Second, we can alleviate the assumption that the $a_i \cdot n$ values (similarly

$-a_i \cdot n)$ are integral for all $i$, by rounding down the value in case it is not integral, and choosing $n - \sum_{i=0}^{k} \lceil a_i \cdot n \rceil$ additional elements in $P$ so that each has probability $1/n$ (and analogously for $Q$). By Claim 5.5 in Raskhodnikova et al. (2009) this process increases the number of values in $P$ and $Q$ by at most $O(k^2)$. Hence, the distributions are now well defined, and we can rely on the following theorem from Raskhodnikova et al. (2009).

**Theorem 3** (Corollary 5.7 in Raskhodnikova et al. (2009), restated.). *Let $P$ and $Q$ be random variables over positive integers $b_1 < \ldots < b_{k-1}$, that have matching moments 1 through $k-1$. Then for any Poission-s algorithm $\mathcal{A}$ that succeeds to distinguish $P$ and $Q$ with high probability, $s = \Omega(n^{1-1/k})$.*

Therefore, plugging the value of $\varepsilon$, we get an $s = \Omega(n^{1-\Theta(\log(1/\varepsilon))})$ lower bound as claimed. $\qquad \square$

### A.3 BOUNDS WHEN THE PREDICTOR IS ACCURATE UP TO LOW TOTAL VARIATION DISTANCE

In this section, we consider estimating support size when we are promised a bound on the total variation distance, or $\ell_1$ distance, between the prediction oracle and the true distribution. First, we show that if the predictor is accurate up to extremely low total variation distance, then the algorithm of Canonne & Rubinfeld (2014) is optimal.

**Theorem 4.** *Let $\Pi$ represent our predictor, where for each sample we are given a random sample $i \sim \mathcal{P}$ along with $\Pi(i)$. If we define $p(i) := \mathbb{P}(i \sim \mathcal{P})$, then if $\|\mathcal{P} - \Pi\|_1 = \sum_{i=1}^{N} |p(i) - \Pi(i)| \leq \frac{\varepsilon}{2}$, then using $O(\varepsilon^{-2})$ samples, the algorithm of Canonne and Rubinfeld can approximate the sample size up to an $\varepsilon \cdot n$ additive error.*

*Proof.* The algorithm of Canonne & Rubinfeld (2014) works as follows. For each sample $(i, \Pi(i))$, it computes $1/\Pi(i)$, and averages this over $O(\varepsilon^{-2})$ samples.

From now on, we also make the assumption that $\Pi(i) \geq 1/n$ for all $i$. This is because we know $p(i) \geq 1/n$ whenever $i$ is sampled, so the predictor $\Pi' = \max(\Pi, 1/n)$ is a strict improvement over $\Pi$ for any $i$ that is sampled.

If we draw $i \sim \mathcal{P}$, then $\mathbb{E}[\frac{1}{p(i)}] = \sum_{i:p(i) \neq 0} p(i) \cdot \frac{1}{p(i)} = S$, where $S$ is the support size. Also, $\mathbb{E}[\frac{1}{\Pi(i)} - \frac{1}{p(i)}] = \sum_{i:p(i) \neq 0} p(i) \cdot (\frac{1}{\Pi(i)} - \frac{1}{p(i)}) = \sum[\frac{p(i)}{\Pi(i)} - 1]$, which, if $\Pi(i) \geq \frac{1}{n}$, is bounded in absolute value by $\sum |\frac{p(i)}{\Pi(i)} - 1| \leq n \cdot \sum |p(i) - \Pi(i)| \leq \frac{\varepsilon \cdot n}{2}$. So, $\mathbb{E}_{i \sim \mathcal{P}}[\frac{1}{\Pi(i)}] \in [S - \varepsilon \cdot n, S + \varepsilon \cdot n]$. Since we have made sure that $\Pi(i) \geq \frac{1}{n}$ whenever $x$ is sampled, $Var(\frac{1}{\Pi(i)}) \leq n^2$. Therefore, by a basic application of Chebyshev's inequality, an average of $O(\varepsilon^{-2})$ samples of $\frac{1}{\Pi(i)}$ is sufficient to estimate $S$ up to a $\varepsilon \cdot n$ additive error. $\qquad \square$

The Canonne and Rubinfeld algorithm has been shown to be far less robust to the error of our predictor, even when given an equal number of samples as our main algorithm. This suggests that an $\varepsilon$-total variation distance bound assumption is a less suitable assumption in our context.

Conversely, if the predictor is only promised to be accurate up to not-as-low total variation distance, we show that no algorithm can do significantly better than Wu & Yang (2019).

**Theorem 5.** *Let $\Pi$ represent our predictor, and recall that $p(i) := \mathbb{P}(i \sim \mathcal{P})$. Suppose the only promise we have on $\Pi$ is that $\|\mathcal{P} - \Pi\|_1 = \sum_{i=1}^{N} |p(i) - \Pi(i)| \leq \gamma$ for some $\gamma \geq C\varepsilon$, where $C$ is some sufficiently large fixed constant and $\varepsilon \geq \frac{1}{n}$. Then, there does not exist an algorithm that can learn the support size of $\mathcal{P}$ up to an $\varepsilon \cdot n$ additive error using $o(n/\log n)$ samples.*

*Proof.* Consider some unknown distribution $\mathcal{D}$ over $[n]$ that we want to learn the support size of, with only samples and no frequency predictor. Moreover, assume that for all $i \in [n]$, $D(i) \geq 1/n$ whenever $D(i) > 0$, where $D(i) = \mathbb{P}(i \sim \mathcal{D})$. Now, let $m = n/\gamma$, and consider the following distribution $\mathcal{P}$ on $[m]$. For $n < i \leq m$, let $p(i) = \frac{1}{m} = \frac{\gamma}{n}$, and for $1 \leq i \leq n$, let $p(i) = \gamma \cdot D(i)$. Let $\Pi$ be the uniform distribution on $[n+1 : m]$, i.e., $\Pi(i) = 0$ if $i \leq n$ and $\Pi(i) = \frac{1}{m-n}$ if $i \geq n+1$. It is clear that $\mathcal{P}$ and $\Pi$ are both distributions over $[n]$ (i.e., the sum of the probabilities

equal 1), the minimum nonzero probability of $\mathcal{P}$ is at least $\frac{\gamma}{n} = \frac{1}{m}$, and that $TV(\mathcal{P}, \Pi) \leq \gamma$. Finally, the support size of $\mathcal{P}$ is precisely $m - n$ more than the support size of $\mathcal{D}$.

Assume the theorem we are trying to prove is false. Then, there is an algorithm that, given $k = o(\frac{m}{\log m})$ random samples $(i, \Pi(i))$ where each $i \sim \mathcal{P}$, can determine the support size of $\mathcal{P}$ up to an $\varepsilon \cdot m$ factor. Then, since the support size of $\mathcal{D}$ is exactly $m - n$ less than the support size of $\mathcal{D}$, one can learn the support size of $\mathcal{D}$ up to an $\varepsilon \cdot m \leq \frac{\varepsilon}{\gamma} \cdot n \leq \frac{n}{C}$ additive error using $k$ samples $(i, \Pi(i))$ for $i \sim \mathcal{P}$.

Now, using only $\Theta(\gamma \cdot k)$ samples from $\mathcal{D}$, it is easy to simulate $k$ samples from $(i, \Pi(i))$, where $i \sim \mathcal{P}$. This is because to generate a sample $(i, \Pi(i))$, we first decide if $i > n$ or $i \leq n$ (the latter occurring with probability $\gamma$). In the former case, we draw $i$ uniformly at random and let $\Pi(i) = \frac{1}{m-n}$. In the latter case, we know that $\mathcal{P}$ conditioned on $i \leq n$ has the same distribution as $\mathcal{D}$, so we just draw $i \sim \mathcal{D}$ and $\Pi(i) = 0$, since $\Pi$ is supported only on $[n+1 : m]$. With very high probability (by a simple application of the Chernoff bound), we will not use more than $\Theta(\gamma \cdot k)$ samples from $\mathcal{D}$.

Overall, this means that using only $\Theta(\gamma \cdot k) = o\left(\frac{\gamma \cdot m}{\log m}\right) = o\left(\frac{n}{\log m}\right) = o\left(\frac{n}{\log n}\right)$ samples (since $\gamma \geq \varepsilon \geq 1/n$) from $\mathcal{D}$, we have learned the support size of $\mathcal{D}$ up to an $\frac{n}{C}$ additive error. This algorithm works for an arbitrary $\mathcal{D}$ and the predictor $\Pi$ does not actually depend on $\mathcal{D}$. Therefore, for sufficiently large $C$, this violates the lower bound of Wu & Yang (2019), who show that one cannot learn the support size of $\mathcal{D}$ up to $\frac{n}{C}$ additive error using $o\left(\frac{n}{\log n}\right)$ samples. $\qquad\square$

