# OpenReview forum: "Learning-based Support Estimation in Sublinear Time"
_ICLR.cc/2021/Conference — ICLR 2021 Spotlight_

### Official Review · AnonReviewer3 · 2020-10-15
**Review of "Learning-based Support Estimation in Sublinear Time"**

**Rating:** 7
**Confidence:** 4

**Review:**

This paper considers the support size estimation problem using a random sample from the unknown distribution and access to some predictor of the element frequency. Under that setting, the paper improves the estimator by Wu & Yang (2019) by refining the approximation interval promised by the predicted frequency. A theoretical upper bound on the sample complexity is proved in Theorem 1 using the proposed algorithm, and it is nearly optimal as shown by the lower bound in Theorem 2. The algorithm is empirically evaluated by both real and synthetic datasets. The empirical performance improves existing algorithms of WY and CR in most cases.

The algorithmic design and analysis are based on the Chebyshev polynomials by Wu & Yang (2019). Given access to the estimated element frequency, the proposed algorithm uses a smaller approximation interval that yields a smaller approximation error and thus a smaller bias. The performance improves WY which has no access to any frequency predictor, and also improves CR that requires perfect frequency information. Overall I think this paper is theoretically sound and empirically convincing. I will recommend for acceptance after the concerns below are addressed.

One major concern is the claim of sublinear time in the title. This seems to be not justified in the paper. From my understanding, given a dataset, the computation time consists of the following: train a frequency predictor, incrementally test the base parameter, construct estimators in each interval, collecting samples, call the frequency predictor, compute each estimator and report the sum. I can see some steps are easy to compute (such as computing a linear estimator), but the overall time complexity should be justified.

Another concern is the theoretical guarantee of the frequency predictor. Estimating element frequency seems not an easy problem by itself given such a small amount of samples relative to $n$. The theorem requires $\Pi(i)\le p_i \le b\Pi(i)$ for a constant $b$. Please comment on how realistic is this assumption. I understand this is not the main concern of the current paper, and empirically predictors indeed help. But I think some proper introduction or literature review on the frequency predictor should be included, as this is the major side information for improvement. Minor comment: it might be better to write something like $\Pi(i)/\sqrt{b}\le p_i \le \sqrt{b}\Pi(i)$ which tolerates error in both upper and lower bounds.

In the theory part, some statements should be more rigorous. In Theorem 1, it is unclear what is a precise meaning of "with high probability", and what is the o(1) term with respect to. In Section 2.1, for $N_i> \Omega(\log n)$ with high probability, it seems to require $p_i > \Omega(\log n/N)$. I think by "random sample" this paper refers to a random sample with replacement, so in the last line on page 1, one needs $O(n\log n)$ samples to see all elements (coupon collector problem).

In the experiment part, it seems the base parameter $b$ is pretty large for some datasets (like more than 400 for IP). It seems that the proposed algorithm reduces to WY once $b$ is moderately large such that one interval suffices. Please explain the large values of $b$. Also, please specify how do you increment the base parameter $b$ when the interval fails the sanity check. The improvement over WY is not surprising as WY doesn't have access to the extra predictor, but CR seems to completely fail in some cases which deserves further discussions.

---

> ### Author Response · Authors · 2020-11-17
> **Response to AnonReviewer3**
>
> Thank you for your feedback and comments.
>
> Regarding the claim of sublinear time in the title: First, let us clarify our notion of sublinearity. It refers to sublinearity in $n$, the known upper bound on the support size (this is the standard notion on sublinearity in the literature on this problem). Furthermore, importantly, the sublinearity claim assumes we already have access to a ready-made oracle. In practice, the oracle needs to be trained (or otherwise produced) in advance. This happens in a preprocessing stage, before the input distribution is given to the algorithm, and this stage is not accounted for in the sublinear running time. We also note that training the oracle needs to be done only once for all future inputs (not once per input). We have updated the paper to clarify these points.
>
>
> Now let us specify the steps of the algorithm (given a ready-made oracle) and their runtimes. We draw $N=O(n^{1-1/\log(1/\epsilon)}\cdot\log(1/\epsilon))$ samples. For each of them, we query the oracle for a frequency estimate. In theoretical oracle models (e.g., Cannone-Rubinfeld’14) it is customary to assume that an oracle query takes $O(1)$ time. In practice, for us this means performing inference on the pre-trained RNNs of Hsu et al., which are small 2-layer networks, so inference is indeed fast (and in particular the time per oracle query is independent of $n$). Then, for a given base parameter $b$, we compute an estimator in time $O(N)$. Since we iterate over $O(1)$-many candidates for $b$ (the number of candidates is independent of $N$ and of $n$), the total running time is $O(N)$, which is $o(n)$.
>
> We also remark that the base iteration procedure does not entail additional samples, nor additional oracle calls. The estimators for all base candidates are computed on the same set of samples and frequency predictions. Thus, the total number of samples and oracle calls is exactly $N$, not $O(N)$ with a larger hidden constant.
>
> Regarding the theoretical guarantee of the frequency predictor: The primary motivation for our choice of this assumption is that it seems a natural formal model for ML-based predictors, which are not fully accurate but often come very close. Please see our above answer to AnonReviewer4 for elaboration on this point. More specifically regarding the problem of estimating frequencies: First, we have to assume access to a preprocessing stage, since otherwise we run into the tight Wu-Yang lower bound. If we do assume a preprocessing step that allows only sample access to a distribution over a domain of size $n$, then estimating the frequency of every element up to a constant factor (simultaneously) takes $O(n\log n)$ samples (by Chernoff and a union bound) for each new data set.
> When dealing with multiple related data sets, however, invoking ML mechanisms is beneficial due to their ability to give meaningful information on related data sets, allowing us to prepare (train) an oracle prior to gaining access to the actual input distribution.
>
> Regarding two-sided error: Our results hold without change even if we modify the approximation premise to $\Pi(i)/b’ \leq p_i \leq b"\Pi(i)$ for any $b’,b"$ such that $b’b"=b$. We specifically chose $b’=1$ and $b"=b$ for convenience, as it renders the Chebyshev polynomial analysis cleaner.
>
> Regarding rigorous statements in the theory part: Thank you for the careful reading.
> 1. “With high probability” should be with constant probability arbitrarily close to 1 (say $0.9$ for concreteness).
> 2. The o(1) term is in fact not necessary (can be zero).
> 3. In Section 2.1, having $N_i > \Omega(\log n)$ whp indeed requires $p_i>\Omega(\log n / N)$, this is how WY handle "heavy elements" (whereas light ones are handled by the Chebyshev polynomial estimator). We apologize if we could not understand your meaning in this comment, please inform us if this point remains unclear.
> 4. In the end of page 1, we meant that counting the number of distinct elements in a sample of size $O(n\log(1/\epsilon))$ is sufficient for achieving additive error $\pm\epsilon n$ with probability (say) $0.9$ (that paragraph suppresses the runtime dependence on $\epsilon$ and focuses on $n$). However, it is correct that for exact counting we are in the coupon collector setting and require $\Theta(n\log n)$.
>
> We have corrected and/or clarified the above points in the paper.
>
> Regarding the performance of CR: It is indeed very sensitive to the accuracy of the oracle. It can be shown that CR incurs a very large error (close to $n$) on simple instances, if the frequency oracle is accurate only up to a constant factor bounded away from 1, even if CR is given an unbounded number of samples. Please refer to our above answer to AnonReviewer1 for details.

---

> > ### Author Response · Authors · 2020-11-17
> > **Continued response**
> >
> > Regarding the base parameter: Indeed, in practice it turns out to be rather large, for reasons discussed in Section 3.1 (in brief: (1) larger $b$ accommodates less accurate empirical predictors, (2) smaller $b$ means many intervals with few samples in each, inhibiting the accuracy of the statistical estimate per interval). However, in our experiments it never becomes so large as to be essentially equivalent to using only one interval (which would coincide with WY). Even in its highest setting (in the IP dataset) there are at least two intervals with non-trivially many samples in each (and indeed, the returned estimates are different and better than WY). Regarding the selection procedure: when a sanity check fails, the base parameter is incremented by 1.

---

### Official Review · AnonReviewer4 · 2020-10-24
**Insightful discussion on the new assumption**

**Rating:** 8
**Confidence:** 4

**Review:**

The paper studies the following problem: suppose there is a distribution $P$ over $n$ class {$0, 1, \dots n-1$},  and one has access to a set of samples drawn from $P$. The goal is to estimate $k$, the number of classes whose probability mass is non-zero.

First of all, it seems the literature addresses the problem by assuming each $p_i \geq 1/n$, where $p_i$ is the mass for class $i$. Thus, by Chernoff bound it is not surprising to see that taking $O(n)$ samples suffices. However, recent progress showed that a bound of $O(n / \log n)$ can be achieved.

In this paper, by assuming that there exists some fairly good estimate of $p_i$, i.e. we have $\Pi_i$ on hand with $1 \leq \Pi_i / p_i \leq b$ for some known constant $b$, authors show that the above sample complexity can be further improved to $\log(1/\epsilon) \cdot \frac{n}{n^{\log(1/\epsilon)}}$.

My major concern is such assumptions might be too strong.
- Authors simply cite couple of related works and say: (A1) $p_i > 1/n$ is mild condition. I am not convinced by such argument. If we drop such assumption, is there any fundamental challenge to well define the problem?
- On top of it, authors further argue that: (A2) there are good estimates of $p_i$ available to us, and the approximation parameter $b$ is also known. It looks fairly artificial for such two-sided approximation condition to hold, and it is not even of theoretical interest.

On the algorithmic end, it looks interesting to draw a more careful analysis on top of [WY19]. However, the unrealistic assumption (A2) does diminish the contribution.

---

> ### Author Response · Authors · 2020-11-17
> **Response to AnonReviewer4**
>
> Thank you for your feedback and comments.
>
> Regarding assumption (A1): Indeed, this is a common assumption throughout the long line of past research on this problem. The justification for it is twofold.
>
> First, without a lower bound on the minimum non-zero probability, no support estimation algorithm is possible (even if the number of samples is allowed to be an arbitrarily large function of $n$ -- i.e., not only sublinear algorithms are impossible). The reason is that there could be arbitrarily many supported elements with arbitrarily small probability. See for example the discussion in Orlitsky et al. 2016, supplementary, Section 5 (https://www.pnas.org/content/pnas/suppl/2016/11/07/1607774113.DCSupplemental/pnas.1607774113.sapp.pdf).
>
> Second, one of the main applications of the support estimation problem is estimating the number of different classes in a population (we refer for example to the hundreds of references collected in https://courses.cit.cornell.edu/jab18/bibliography.html), a.k.a, the Distinct Elements problem. This problem is reduced to support size estimation by defining a distribution over the classes, where the probability mass of each class is its (unknown) number of occurrences in the population divided by the population size $n$. Since each class appears in the population a non-negative integer number of times, the minimum probability mass of any occurring class is at least $1/n$.
>
> We apologize for not being clear on the justifications for this assumption and have elaborated on them in the paper. The discussion about the connection to the distinct elements problem has been included in the original submission (see footnote 2 on page 1).
>
> Regarding assumption (A2): In general, the motivation for considering models with approximate oracles is the recently emerging research effort to harness advances in machine learning to develop better algorithms. While learning mechanisms like neural networks cannot be assumed to return accurate numerical predictions, they can often provide approximations, so approximate oracles are a natural choice for developing new algorithms. Such oracle models have recently become prevalent and successful, see for example the survey of Mitzenmacher and Vassilvitskii 2020 (https://arxiv.org/pdf/2006.09123.pdf).
>
> In order to formally reason about ML predictors as oracles for an algorithm, it is necessary to make a modeling assumption about them. While it is certainly true that learning mechanisms such as RNNs do not guarantee a constant-factor two-sided approximation, an equally important question is whether such a modeling choice is useful in gaining insight into the problem (as per the famous saying of statistics: “all models are wrong, but some are useful”). We deem this model a natural choice, and it indeed leads us to a new algorithm which is shown to outperform the state of the art in practice, when invoked with the pre-trained RNN frequency predictors of Hsu et al. (2019). While we believe that the approximate frequency setting is theoretically interesting on its own right, our main point in this paper is that it also leads to useful practical implications.
>
> Regarding the assumption that $b$ is also known: The main theorem and the basic algorithm (Algorithm 1) indeed assume that $b$ is known. However, we emphasize that in our experiments, no such assumption is needed: we develop a method to select $b$ in Section 3.1, and study its behavior (Figures 1,3,5) and performance (Figures 2,4,6) on our datasets.

---

> > ### Comment · AnonReviewer4 · 2020-11-17
> > **response**
> >
> > Thanks for the response. Your revision does clarify why (A1) is vital for the problem of interest.
> >
> > However, I am still not convinced why "the approximate frequency setting is theoretically interesting on its own right". As discussed in my initial review, such point-wise two-sided approximation assumption is too strong to hold. A more practical alternative is to consider a distribution Q whose TV distance to P is upper bounded by some parameter. Does this translate into similar conclusion as in the current paper?

---

> > > ### Author Response · Authors · 2020-11-19
> > > **Response**
> > >
> > > We thank the reviewer for raising this interesting point in our choice of model.
> > >  We managed to show that if we only assume that the predictor outputs values according to some distribution that is close in TVD, then we can only get an estimator with an error of $\pm O(\epsilon n)$  if the total variation distance is at most $O(\epsilon)$ (see below for the proof). However, if we had a predictor satisfying such a strong guarantee, then the prior algorithm by Canonne and Rubinfeld would work as well (the proof is also included below), while our experiments shows that it suffers from a large error on multiple data sets. Thus, we believe that our assumption (a pointwise approximation guarantee, but with an arbitrary constant multiplicative approximation factor) is better suited for our application.
> > >
> > > **$\gamma$-TVD closeness for $\gamma>C\varepsilon$ is not sufficient.**
> > > Let $P$ be some unknown distribution over $[m] = \{1, 2, \dots, m\}$, such that if $P(x) > 0$ then $P(x) > 1/m$. For each $P$, let $Q$ be some distribution such that $TV(P, Q) \le \gamma$, for some parameter $\gamma > C \varepsilon$ for $C>1$. We think of $Q$ as the distribution according to which the neural network (frequency predictor) replies.
> > > Suppose there is an algorithm that, given $k = o(\frac{m}{\log m})$ random samples $(x_i, q(x_i))$ where each $x_i \sim P$, can determine the support size of $P$ up to an $\varepsilon \cdot m$ factor. Then, one can learn the support size of a distribution $D$ over $[n]$ up to error $n/C$ using $o(\frac{n}{\log n})$ samples without access to the $q(x_i)$ values, if $m = n/\gamma$. This contradicts the lower bound of Wu and Yang for some fixed constant $C$. Therefore, the neural network would need to learn $P$ up to total variation distance $O(\varepsilon)$.
> > >
> > >
> > > To prove this, consider some unknown distribution $D$ over $[n]$ that we want to learn the support size of, with only samples and no frequency predictor. We assume $D(x) \ge 1/n$ if $D(x) > 0$ for all $x$. Consider the following distribution $P$ on $[n/\gamma]$. For $n < x \le n/\gamma,$ let $P(x) = \frac{\gamma}{n},$ and for $1 \le x \le n,$ let $P(x) = \gamma \cdot D(x)$.  Let $Q$ be the uniform distribution on $[n+1: n/\gamma]$. It is clear that the minimum nonzero probability of $P$ is at least $\frac{\gamma}{n} = \frac{1}{n/\gamma},$ and that $TV(P, Q) \le \gamma$.
> > >
> > > Now, if we are given $\Theta(k)$ samples from $D$, it is easy to simulate $k/\gamma$ samples from $(x_i, q(x_i)),$ where $x_i$ is distributed as $P$. This is because to generate a sample $(x_i, q(x_i)),$ we first decide if $x > n$ or $x \le n$ (the latter occurring with probability $\gamma$). In the former case, we draw $x$ uniformly at random and let $q(x_i) = \frac{1}{n/\gamma - n}.$ In the latter case, we know that $P$ conditioned on $x \le n$ is the same as $D$, so we just draw $x_i \sim D$ (or just use the next sample from $D$ we are given) and $q(x_i) = 0$, since $Q$ is supported only on $[n+1: n/\gamma]$. With very high probability (by Chernoff), we will not use more than $\Theta(k)$ samples from $D.$
> > >
> > > Now, we can use the algorithm we are promised, that estimates the support size of $P$ up to an $\varepsilon \cdot m$ factor, where $m = n/\gamma$, as long as $k/\gamma = o(\frac{m}{\log m})$. But $\gamma > C \varepsilon,$ so $\varepsilon \cdot m \le n/C.$ Since the support size of $P$ is just $(m-n)$ more than the support size of $D$, we can establish the support size of $D$ up to an $\varepsilon \cdot n$ error without any help from the predictor, using only $o(\gamma \cdot \frac{m}{\log m}) = o(\frac{n}{\log n/\gamma}) = o(n/\log n)$ samples from $D$.

---

> > > > ### Author Response · Authors · 2020-11-19
> > > > **Continued response**
> > > >
> > > > **$\gamma$-TVD closeness for $\gamma<\varepsilon$ suffices for Canonne-Rubinfeld algorithm.**
> > > >  If the learned distribution has $TV(P, Q) \le \varepsilon$, then using $O(\varepsilon^{-2})$ samples we can approximate the sample size up to a $2 \cdot\varepsilon \cdot n$ factor, using algorithm of Canonne and Rubinfeld. The algorithm of Canonne and Rubinfeld works as follows. For each sample $(x, q(x)),$ where $q$ is the learned probability, it computes $1/q(x)$, and averages this over $O(\varepsilon^{-2})$ samples.
> > > >
> > > > To see why it can be done in $O(\varepsilon^{-2})$ samples, if we draw $x \sim P$, $\mathbb{E}[\frac{1}{p(x)}] = \sum_{x: p(x) \neq 0} p(x) \cdot \frac{1}{p(x)} = S$, where $S$ is the support size. Also, $\mathbb{E}[\frac{1}{q(x)}-\frac{1}{p(x)}] = \sum_{x: p(x) \neq 0} p(x) \cdot (\frac{1}{q(x)}-\frac{1}{p(x)}) = \sum [\frac{p(x)}{q(x)} - 1],$ which, if $q(x) > \frac{1}{n},$ is bounded in absolute value by $\sum |\frac{p(x)}{q(x)} - 1| \le n \cdot \sum |p(x)-q(x)| \le \varepsilon \cdot n$. The assumption that $q(x) \ge \frac{1}{n}$ for every $x$ sampled isn't necessarily true. However, once $x$ is sampled, we know that the true probability $p(x) \ge \frac{1}{n},$ so if $q(x) < \frac{1}{n}$ (which may include the case $q(x) = 0$), we can replace $q(x)$ with $\frac{1}{n},$ and this only reduces the total variation distance. So, $\mathbb{E}_p[\frac{1}{q(x)}] \in [S - \varepsilon \cdot n, S + \varepsilon \cdot n]$. Since we have made sure that $q(x) \ge \frac{1}{n}$ whenever $x$ is sampled, $Var(\frac{1}{q(x)}) \le n^2$. Therefore, by a basic application of Chebyshev's inequality, an average of $O(\varepsilon^{-2})$ samples is sufficient to estimate $S$ up to a $2 \varepsilon \cdot n$ additive error.
> > > >
> > > >
> > > > The Canonne and Rubinfeld algorithm has been shown to be far less robust to the error of our predictor, even when given an equal number of samples as our main algorithm. This suggests that a $\varepsilon$ total variation distance is a less suitable assumption in our context.

---

> > > > > ### Comment · AnonReviewer4 · 2020-11-19
> > > > > **Response to the concrete discussion on (A2)**
> > > > >
> > > > > Thanks for the detailed discussion on the gain and limits of using TV distance as alternative condition. I believe while TV does not lead to as optimal results as concluded in the paper, authors did a great work in clarifying what can be expected from a weaker condition and why (A2), i.e. a stronger condition on initial state, might be vital for the problem being studied. I feel this complements perfectly with the positive results already been carried out, and they together attribute to a clear acceptance.

---

### Official Review · AnonReviewer2 · 2020-10-28
**Nice blend of theory and practice**

**Rating:** 8
**Confidence:** 3

**Review:**

Estimation of the size of the support of a distribution over a discreet domain is a fundamental problem. In the standard setting, this problem is theoretically well-understood with matching upper and lower bounds. The authors assume additional access to a constant approximation of the density function at each point, and then show that this can provably reduces the sample complexity. In particular, they offer matching upper and lower bounds in this setting. While the upper bound is a twist on the existing state-of-the-art method, the lower bound seem to deviate from that.

One may say that the problem of density estimation is harder than estimating the size of the support of a distribution, and therefore assuming access to such oracle is not natural. However, the authors provide practical evidence that in some cases this is reasonable. In particular, the authors use a learning-based method for estimation of the density function and plug it in as the oracle for their support estimation method. The results are promising, and seem to be more robust than a previous method that assumed access to an accurate pdf oracle.

It will be quite interesting to discuss the kind of structure that is present in the data sets that allows to improve over the WY method. For example, is it the case that WY method performs poorly compared to the proposed method for light-tails distributions (and not so much for heavy tailed distributions)?

The paper is well written and the background work is adequately discussed.

For some reason the submission format does not allow selecting/highlighting text in the pdf file. Please check this.

---

> ### Author Response · Authors · 2020-11-17
> **Response to AnonReviewer2**
>
> Thank you for your feedback and comments.
>
> Regarding properties present in the data that allow for improvement by learning-based techniques: our understanding of this issue is incomplete, but we believe that the quality of the prediction (i.e., how well the predictor approximates the actual distribution density) is more important than the distribution itself. In particular, if the prediction is perfect, the algorithm of Canonne-Rubinfeld returns a good estimation using only a constant (independent of n) number of samples, irrespectively of the input distribution. That said, it is possible that obtaining good predictors is easier for some distributions than others. We will explore this question in future work.
>
>
> Regarding the selecting/highlighting issue: we apologize for this technical issue, we had not noticed it before. We believe it is fixed now.

---

### Official Review · AnonReviewer1 · 2020-11-04
**Interesting learning approach for a classical statistical problem**

**Rating:** 7
**Confidence:** 4

**Review:**

The paper considers the problem of estimating the support of a discrete distribution, when provided access to samples and an oracle that approximately predicts the probability of the observed sample.

They propose an algorithm based on Chebyshev polynomials  and also show that the proposed algorithm is optimal. They evaluate the algorithm on two public datasets and a synthetic dataset and show that the algorithm performs reasonably well. The results are interesting and I recommend acceptance.

The main technical contribution is to use the approximate probability of the sample to divide the interval [0, log n /n] into exponential bins and use the best Chebyshev approximation within each interval.
I strongly encourage authors to add technical comparisons between their work and that of Canonne and Rubinfeld 2014 and other relevant papers e.g.,
1. Optimal Bounds for Estimating Entropy with PMF Queries
2. Probability–Revealing Samples

I am also curious to know if similar results hold for unseen species estimation (e.g., https://www.pnas.org/content/113/47/13283.short)?

---

> ### Author Response · Authors · 2020-11-17
> **Response to AnonReviewer1**
>
> Thank you for your feedback and comments.
>
> Regarding the comparison to previous work:
>
> The CR algorithm samples $m$ elements $i_1,...,i_m$ and returns the average of their inverse masses, $\tfrac1m\sum_{j=1}^mp_{i_j}^{-1}$. It can be seen to be sensitive to the precision of its access to the masses $p_{i_j}$. For example, suppose the underlying distribution is uniform over $\{1...n\}$, and we have a $b$-approximate oracle (for some $b=O(1)$), that reports the mass of each element as $b/n$ instead of $1/n$. Then, the CR algorithm returns $n/b$ regardless of the number of samples. The additive error is thus $(1-1/b)n$, much larger than our additive error of $\epsilon n$.
>
> The PMF paper: Thank you for pointing out this highly relevant work. We have now included it in the revised paper. Indeed, they consider a similar model to ours, where the algorithm only has access to an approximation of each $p_i$ rather than its exact value. However, they assume a much finer approximation, namely a $(1+\epsilon/3)$-approximation, whereas we need only assume a constant approximation with any constant. Their algorithm is in fact the same as the CR algorithm; their novelty is in showing it can in fact tolerate a $(1+\epsilon/3)$-approximation of the $p_i$’s. However, it cannot tolerate larger constant approximations, as demonstrated above. We have included the above discussion in the revised version of the paper.
>
> Regarding possible connections to unseen species estimation: This is a very interesting direction -- indeed, Section 5 in the supplementary material of the paper you cite is dedicated to discussing connections and differences between unseen species estimation and support size estimation. There are certain key differences between the two problems, and whether our techniques can be extended to the former ones requires further exploration.

---

> > ### Comment · AnonReviewer1 · 2020-11-20
> > **response**
> >
> > Thanks for the clarifications. If accepted, please include these detailed comparisons in the final version.

---

### Author Response · Authors · 2020-11-17
**Response to reviews**

We thank the reviewers for their valuable feedback. Answers are given in a response to each review. We have uploaded a revised version of the paper, with newly added segments marked in blue for the convenience of the reviewers (we will remove the coloring in the final version).

---

### Decision · Program_Chairs · 2021-01-07
**Final Decision**

**Decision:**

Accept (Spotlight)

**Comment:**

The paper gives a learning-augmented algorithm for estimating the support size of a discrete distribution. The proposed algorithm is evaluated experimentally, showing significant improvements in the estimation accuracy. The reviewers unanimously agreed that the contributions are strong and relevant. I recommend accept.